# Assessment of antibiotic prescribing pattern and cost for hospitalized patients: A study from Palestine

**Rufayda Dawood Manassrah**[1]*, **Rowa Al Ramahi**[2]

**1** Faculty of Graduate Studies, Department of Pharmacy, An-Najah National University, Nablus, Palestine,
**2** Faculty of Medicine and Health Sciences, Department of Pharmacy, An-Najah National University, Nablus, Palestine

* rufidamanassrah@yahoo.com

**Data Availability Statement:** All relevant data are within the manuscript and its Supporting Information files.

**Funding:** The author(s) received no specific funding for this work.

## Abstract

### Background

One of the largest problems facing the world today is the morbidity and mortality caused by antibiotic resistance in bacterial infections. A major factor in antimicrobial resistance (AMR) is the irrational use of antibiotics. The objective of this study was to assess the prescribing pattern and cost of antibiotics in two major governmental hospitals in the West Bank of Palestine.

### Methods

A retrospective cohort study was conducted on 428 inpatient prescriptions containing antibiotics from two major governmental hospitals, they were evaluated by some drug use indicators. The cost of antibiotics in these prescriptions was calculated based on the local cost. Descriptive statistics were performed using IBM-SPSS version 21.

### Results

The mean ± SD number of drugs per prescription (NDPP) was 6.72 ± 4.37. Of these medicines, 38.9% were antibiotics. The mean ± SD number of antibiotics per prescription (NAPP) was 2.61 ± 1.54. The average ± SD cost per prescription (CPP) was 392 ± 744 USD. The average ± SD antibiotic cost per prescription (ACPP) was 276 ± 553 USD. The most commonly prescribed antibiotics were ceftriaxone (52.8%), metronidazole (24.8%), and vancomycin (21.0%). About 19% of the antibiotics were prescribed for intra-abdominal infections; followed by 16% used as prophylactics to prevent infections. Almost all antibiotics prescribed were administered intravenously (IV) 94.63%. In general, the average duration of antibiotic therapy was 7.33 ± 8.19 days. The study indicated that the number of antibiotics per prescription was statistically different between the hospitals ($p = 0.022$), and it was also affected by other variables like the diagnosis ($p = 0.006$), the duration of hospitalization ($p < 0.001$), and the NDPP ($p < 0.001$). The most commonly prescribed antibiotics and the cost

**Competing interests:** The authors have declared
that no competing interests exist.

of antibiotics per prescription were significantly different between the two hospitals (p <
0.001); The cost was much higher in the Palestinian Medical Complex.

## Conclusion

The practice of prescribing antibiotics in Palestine's public hospitals may be unnecessary
and expensive. This has to be improved through education, adherence to recommenda-
tions, yearly immunization, and stewardship programs; intra-abdominal infections were the
most commonly seen infection in inpatients and ceftriaxone was the most frequently admin-
istered antibiotic.

## Introduction

One of the most serious issues confronting the world today is the massive morbidity and mor-
tality caused by antibiotic resistance in bacterial diseases [1]. A major factor in antimicrobial
resistance (AMR) is the irrational use of antibiotics [2]. Understanding the primary causes of
irrational antibiotic use is essential. These include a lack of information and awareness among
the general people, unrestricted access to medicines, leftover antibiotics from previous pre-
scriptions, insufficiently quick and accurate diagnostic procedures, a lack of local antibiotic
susceptibility data, and pharmaceutical promotion; and insufficient training for prescribers,
pharmacists, and medical professionals in terms of their understanding, attitudes, and prac-
tices regarding the use of antibiotics [3]. Recent studies indicate that the COVID-19 pandemic
has made AMR worse [4,5]. There is widespread agreement that the key cause driving AMR
development is antibiotic overuse [6].

The World Health Organization (WHO) developed and implemented strategies to control
irrational medication use, such as drafting and implementing standard treatment guidelines
(STGs) for common illnesses and using essential medicine lists (EMLs) to guide purchase and
training. To ascertain whether or not the use of antibiotics is reasonable, the percentage of
antibiotic use has been considered as a critical indicator [7]. Also, the cost is significant and is
taken into consideration when evaluating the performance of the practitioners' rational use of
drugs, which raises concerns regarding the extent of misalignment between individual and
societal antibiotic costs [2,8].

By 2050, it is anticipated that the number of drug-resistant infection-related deaths will rise
from the current 700,000 to 10 million per year, with a global economic impact of up to 100
trillion US dollars. Therefore, it is possible that the world could soon enter a "post-antibiotic
era" in which common infections cannot be cured [9,10].

Worldwide, the number of deaths could reach 700,000. Furthermore, it is predicted that
multidrug-resistant bacteria cause 25,000 fatalities and 1.5 billion euros in additional health-
care costs and lost productivity per year in Europe [11]. In the United States (U.S.), the Centers
for Disease Control and Prevention (CDC) released the first Antimicrobial Resistance Threats
Report in 2019, raising awareness about the danger posed by antimicrobial resistance. The
report noted that more than 2.8 million infections with antibiotic resistance take place in the
U.S. every year, and as a result, more than 35,000 people die [12].

Antimicrobial resistance is a multi-faceted issue that impacts practically all populations and
is caused by a number of interconnected variables. Single solitary attempts or interventions
only have a small impact. As a result, as part of an international effort, numerous organizations
have established strategies to reduce ambulatory antibiotic consumption [13].

However, the emergence and spread of antimicrobial resistance must be stopped in order to improve the prescribing and use of antibiotics. More coordinated strategies and activities are needed from different stakeholders globally. Governments, agencies, academics, the pharmaceutical industry, healthcare professions, and the community must collaborate in order to achieve this goal via educational, managerial, regulatory, and economic techniques [14].

A good number of previous studies related to antibiotics in our country can be found. However, to the best of our knowledge, no previous studies regarding antibiotic utilization and cost in hospitals are available in the West Bank. Overuse and misuse of these medications can lead to antibiotic resistance, more side effects in patients, and extra cost. Therefore, a follow-up study is required to look at current prescribing habits and then assess the financial impact and justification for this prescribing. For practitioners and policymakers, this might be helpful. Based on these findings, training and instructional programs could be created.

This study aimed to assess the prescribing pattern and cost of antibiotics in two major governmental hospitals in the West Bank of Palestine.

## Methods

### Study setting

The majority of Palestinians residing in the West Bank are entitled to governmental healthcare provided by a network of primary healthcare centers and hospitals. Patients are evaluated and treated by a general practitioner or a specialist. All prescription orders are computerized and kept in the Avicenna Health Information System (HIS). The prescription orders have sections that the prescribing physician must completely fill out. These components include the prescription's origin, the prescriber's name, the patient's information, the patient's current diagnosis, a list of prescriptions with directions for the patient and pharmacist, etc.

The target hospitals were two major governmental hospitals; The Palestinian Medical Complex in Ramallah and Princess Alia Governmental Hospital in Hebron. These areas have a population of 1,137,400 people, making up 21.7% of the population of Palestine [15].

Ethical approval was provided by An-Najah National University Institutional Review Board (IRB) and the Palestinian Ministry of Health before the study. The informed consent was waived by An-Najah IRB.

### Inclusion and exclusion criteria

Only inpatient prescriptions that included antibiotics for any reason were collected from the computerized system (Avicenna HIS) in the included hospitals over a period of four months (1 December 2020–31 March 2021). Convenient sampling was used to collect data.

### Sample size

The estimated sample size using the automated software program, Raosoft sample size program, was 377 prescriptions with more than 20,000 prescriptions as a population size to achieve a confidence level of 95% and a margin of error of 5%. A total of 428 prescriptions containing antibiotics were collected and analyzed retrospectively and systemically.

### Data collection

Considering the importance of data standardization for the internal validity of a study, data collection was standardized by using a Data Collection Form to gather information from Avicenna HIS.

The collected data included sociodemographic data, reason for admission and diagnosis, medications prescribed during hospitalization and duration of hospitalization. They were assessed for the number of drugs per prescription (NDPP), the number of antibiotics per prescription (NAPP), cost per prescription (CPP), antibiotic cost per prescription (ACPP), the cost of medications was evaluated according to the price list available at the site of the Palestinian Ministry of Health, and presented in new Israel shekels (ILS) (ILS 1 = USD 0.29 at the time of study analysis). In addition, main groups of all drugs written on the prescriptions, groups of antibiotics, the most frequently prescribed antibiotics were evaluated.

Antibiotics were grouped by the ATC (Anatomic Therapeutic Chemical) classification. The ATC system is a multi-label classification system developed by the WHO that divides medications into classes based on their properties and therapeutic effects [16].

## Statistical analysis

The prescriptions were carefully analyzed and computerized into the Statistical Package for Social Sciences (IBM-SPSS version 21). A 0 or 1 coding system was used to record data. For each variable, a score of 1 was entered when the variable was present and compliant with the standard. Means ± standard deviations were computed for continuous data. Frequencies and percentages were calculated for categorical variables. Chi-Square and Mann-Whitney tests were used in the statistical analyses. If $p < 0.05$, the comparisons were considered statistically significant.

## Results

### Sociodemographic data

In this study, a total of 428 prescriptions were collected; 242 (56.5%) prescriptions from Princess Alia Governmental (Hebron) Hospital and 186 (43.5%) prescriptions from the Palestinian Medical Complex. The mean age of the patients was 40.45 ± 25.47 years, with 104 years as a maximum and 0.1 years as a minimum, and almost the percentages of gender were equal, male (51.6%) and female (48.8%).

The prescriptions were written mostly by specialists in internal medicine (21.3%), followed by general practitioners (GPs) (19.4%), surgical medicine specialists (18.0%), gynecologists (10.5%), orthopedic and pediatric specialists each provided 9.3% of the prescriptions, urologists (6.8%), neurologists (3.0%), otolaryngologists (ENT) (1.9%), and ophthalmologists (0.5%).

### Assessment of antibiotic use

A total of 2875 medicines were prescribed. The number of drugs per prescription (NDPP) was 6.72 ± 4.37. Of these medicines, 38.9% were antibiotics. The number of antibiotics per prescription (NAPP) was 2.61 ± 1.54.

The common diagnoses and reasons as written in prescriptions were: Intra-abdominal and cardiac infections (18.9%) were the most common, followed by using antibiotics as prophylactic to prevent infections (16.1%), labor or gynecological infections (12.6%), COVID-19 (the SARS coronavirus) (12.4%), lower respiratory tract infections (9.6%), urinary tract infections (8.6%), (osteomyelitis and arthritis) and upper respiratory tract infections each provided (4.4%), and traumatic skin and soft tissue infections (4.0%).

The most commonly prescribed antibiotics were ceftriaxone (52.8%), metronidazole (24.8%), vancomycin (21.0%), and amoxicillin/clavulanic acid (16.6%) (Table 1).

**Table 1. Distribution of the prescribed antibiotics.**

| Antibiotic (ATC Code) | Frequency (n = 428) | Percentage (%) |
|---|---|---|
| Ceftriaxone (J01DD04) | 226 | 52.8 |
| Metronidazole (J01XD01) | 106 | 24.8 |
| Vancomycin (J01XA01) | 90 | 21.0 |
| Amoxicillin + Clavulanic acid (J01CR02) | 71 | 16.6 |
| Cefuroxime (J01DC02) | 66 | 15.4 |
| Azithromycin (J01FA10) | 65 | 15.2 |
| Meropenem (J01DH02) | 65 | 15.2 |
| Ceftazidime (J01DD03) | 51 | 11.9 |
| Ciprofloxacin (J01MA02) | 47 | 11.0 |
| Cefazoline (J01DB04) | 42 | 9.8 |
| Gentamicin (J01GB03) | 40 | 9.3 |
| Piperacillin + Tazobactam (J01CR05) | 40 | 9.3 |
| Ampicillin (J01CA01) | 31 | 7.2 |
| Fucidic Acid (J01XC01) | 28 | 6.5 |
| Teicoplanin (J01XA02) | 19 | 4.4 |
| Colistin (J01XB01) | 15 | 3.5 |
| Levofloxacin (J01MA12) | 15 | 3.5 |
| Sulphamethoxazole + Trimethoprim (J01EE01) | 15 | 3.5 |
| Cefotaxime (J01DD01) | 14 | 3.3 |
| Clindamycin (J01FF01) | 12 | 2.8 |
| Cephalexin (J01DB01) | 11 | 2.6 |
| Amikacin (J01GB06) | 10 | 2.3 |
| Erythromycin (J01FA01) | 9 | 2.1 |
| Doxycycline (J01AA02) | 6 | 1.4 |
| Cloxacillin (J01CF02) | 5 | 1.2 |
| Gemifloxacin (J01MA15) | 4 | 0.9 |
| Rifampicin (J04AM05) | 4 | 0.9 |
| Amoxicillin (J01CA04) | 1 | 0.2 |

Antimicrobials were classified according to the Anatomic Therapeutic Chemical (ATC) classification developed by the WHO (https://www.whocc.no/atc_ddd_index/). When the ATC group distributions of all antibiotics were analyzed, "other beta-lactam antibacterials, the cephalosporins" (ATC code: J01D; 42.87%) were the most frequently prescribed antibiotics, followed by "other antibacterials" (J01X; 23.29%), "beta-lactam antibacterials, penicillins" (J01C; 13.36%), and "macrolides, lincosamides and streptogramins" (J01F; 7.76%).

## Cost analysis of medications and antibiotics

The total cost of all prescribed medicines within the four months was 570,918 ILS (167,471 USD). The average cost per prescription (CPP) was 1,333 ± 2,539 ILS (392 ± 744 USD) with a range from 25 to 22,357 ILS. The total cost of all prescribed antibiotics was 402,693 ILS (118,124 USD), which constituted 70.53% of the total cost of all prescribed drugs. The average antibiotic cost per prescription (ACPP) was 940 ± 1,888 ILS (276 ± 553 USD) with a range from 9 to 19,030 ILS.

Almost all antibiotics prescribed for inpatients were administered intravenously (IV) 94.63% (n = 405), the second route of administration was orally 5.14% (n = 22) and the last route of administration was topically 0.23% (n = 1).

The average duration of hospitalization and antibiotic use was 7.33 ± 8.19 days, with a maximum duration of 55 days and a minimum duration of one day.

## Association between antibiotic number and other variables

All prescriptions written out for inpatients were analyzed in terms of the number of antibiotics and divided into two groups: those with one or two antibiotics (60.9%) and those with three or more antibiotics (39.1%).

Accordingly, the percentage of one or two antibiotics per prescription at Alia Hospital was 60.9%, and 39.1% at the Palestinian Medical Complex. The percentage of three or more antibiotics per prescription at Alia Hospital was 49.7% and 50.3% at the Palestinian Medical Complex. In the comparisons of the NAPP, statistically significant differences were found between the two hospitals ($p = 0.022$).

When the NAPP was examined in terms of the physicians' specialties, the results were found as illustrated in Table 2. No statistically significant differences were found in the comparisons of the number of antibiotics per prescription with the physicians' specialties ($p = 0.116$).

The comparison of the patients' age with the number of antibiotics per prescription did not reveal any statistically significant difference ($p = 0.257$). The percentage of one or two antibiotics per prescription when patients' age was less than 18 years was 21.8%, when the age was between 18 and 65 years it was 60.2%, and when the age was more than 65 years it was 18.0%. The percentage of three or more antibiotics per prescription. The percentage of three or more antibiotics per prescription when patients' age was less than 18 years was 21.0%, when the age was between 18 and 65 years it was 54.5%, and when the age was more than 65 years it was 24.6%.

Also, the comparison of the patients' gender with the number of antibiotics per prescription did not reveal any statistically significant difference ($p = 0.522$). The male percentage of one or two antibiotics per prescription was 52.9%, and when female, it was 47.1%. The male percentage of three or more antibiotics was 49.7%, and when female, it was 50.3%.

In comparisons of the diagnosis on prescriptions with the number of antibiotics per prescription (NAPP), statistically significant differences were found ($p = 0.006$), which means the diagnosis affects the NAPP.

**Table 2. Distribution of the number of antibiotics per prescription (NAPP) with physicians' specialties.**

| | | Number of antibiotics per prescription (NAPP) | | |
| --- | --- | --- | --- | --- |
| | | **One or two antibiotics** | **Three or more antibiotics** | **Total** |
| | | **Frequency(%)** | **Frequency(%)** | **Frequency(%)** |
| **Physicians' specialties** | Internal medicine | 46(17.6%) | 45(26.9%) | 91(21.3%) |
| | GPs | 55(21.1%) | 28(16.8%) | 83(19.4%) |
| | General surgery | 47(18.0%) | 30(18.0%) | 77(18.0%) |
| | Gynecology | 30(11.5%) | 15(9.0%) | 45(10.5%) |
| | Pediatrics | 19(7.3%) | 21(12.6%) | 40(9.3%) |
| | Orthopedics | 29(11.1%) | 11(6.6%) | 40(9.3%) |
| | Urology | 21(8.0%) | 8(4.8%) | 29(6.8%) |
| | Neurology | 7(2.7%) | 6(3.6%) | 13(3.0%) |
| | ENT | 5(1.9%) | 3(1.8%) | 8(1.9%) |
| | Ophthalmology | 2(0.8%) | 0(0.0%) | 2(0.5%) |
| | Total | 261(100.0%) | 167(100.0%) | 428(100.0%) |

**Table 3. Distribution of the number of antibiotics per prescription (NAPP) with the diagnosis on prescriptions.**

| | | Number of antibiotics per prescription (NAPP) | | |
| --- | --- | --- | --- | --- |
| | | **One or two antibiotics** | **Three or more antibiotics** | **Total** |
| | | **Frequency(%)** | **Frequency(%)** | **Frequency%)** |
| **Diagnosis on Prescriptions** | Intra-abdominal infections | 45(17.5%) | 36(21.6%) | 81(18.9%) |
| | Prophylaxis for surgical procedures | 50(19.2%) | 19(11.4%) | 69(16.1%) |
| | Labor and gynecological infections | 37(14.2%) | 17(10.2%) | 54(12.6%) |
| | COVID-19 (SARS coronavirus) | 27(10.3%) | 26(15.6%) | 53(12.4%) |
| | Lower respiratory tract infections | 23(8.8%) | 18(10.8%) | 41(9.6%) |
| | Urinary tract infections | 26(10.0%) | 11(6.6%) | 37(8.6%) |
| | Osteomyelitis and arthritis | 14(5.4%) | 5(3.0%) | 19(4.4%) |
| | Upper respiratory tract infections | 14(5.4%) | 5(3.0%) | 19(4.4%) |
| | Traumatic skin and soft tissue infections | 4(1.5%) | 13(7.8%) | 17(4.0%) |
| | Others | 21(8.0%) | 17(10.2%) | 38(8.9%) |
| | Total | 261(100.0%) | 167(100.0%) | 428(100.0%) |

Intra-abdominal infections and COVID-19 (SARS coronavirus) had a higher percentage of prescriptions that contained three or more antibiotics, while prophylaxis for surgical procedures and labor and gynecological infections had a higher percentage of prescriptions that contained one or two antibiotics (Table 3).

When the durations of antibiotic therapy were analyzed in terms of the number of antibiotics per prescription (NAPP), statistically significant difference was found in the comparison of the number of antibiotics per prescription with the durations of antibiotic therapy ($p < 0.001$)

About four-quarters (73.9%) of prescriptions containing one or two antibiotics were written when the hospitalization duration was five days or less. While two-thirds (66.5%) of prescriptions containing three or more antibiotics were written when the hospitalization duration was six days or more, this suggests that more antibiotics were used when patients were hospitalized for longer periods of time.

The last comparison was between the number of drugs per prescription (NDPP) and the number of antibiotics per prescription (NAPP). Statistically significant differences were found ($p < 0.001$), which means the NDPP affects the NAPP.

Accordingly, about two-thirds (66.7%) of prescriptions containing one or two antibiotics were written when the number of drugs per prescription (NDPP) was five or less. When the NDPP prescribed six or more medications, about 83% of prescriptions containing three or more antibiotics were written. This shows that as the number of medications on the prescription increased, more antibiotics were given out.

## Comparison of infections and antibiotics use between two hospitals

Previously, we compared the number of antibiotics prescribed per prescription and discovered statistically significant differences ($p = 0.022$) between the two hospitals. The percentage of one or two antibiotics per prescription at Alia Hospital was 60.9% and 39.1% at the Palestinian Medical Complex. The percentage of three or more antibiotics per prescription at Alia Hospital was 49.7% and 50.3% at the Palestinian Medical Complex.

During the study period, in a comparison of the written diagnoses on prescriptions, we found that the distribution of the infections in the two hospitals was statistically significantly different ($p < 0.001$). Intra-abdominal and cardiac infections, COVID-19 (the SARS coronavirus), and lower respiratory tract infections were predominant at the Palestinian Medical Complex, and labor or gynecological infections, using antibiotics for surgical prophylaxis, urinary

**Table 4. Distribution of the five most commonly prescribed antibiotics in the two hospitals.**

| Antibiotic (ATC Code) | Alia Hospital | Palestinian Medical Complex | *p* values |
|---|---|---|---|
| Ceftriaxone (J01DD04) | 57.5% | 42.5% | 0.662 |
| Metronidazole (J01XD01) | 52.8% | 47.2% | 0.374 |
| Vancomycin (J01XA01) | 34.3% | 65.6% | < 0.001 |
| Amoxicillin + Clavulanic acid (J01CR02) | 69.0% | 31.0% | 0.020 |
| Cefuroxime (J01DC02) | 68.2% | 31.8% | 0.038 |

tract infections, osteomyelitis and arthritis, upper respiratory tract infections, and traumatic skin and soft tissue infections were predominant at Alia Hospital.

The most frequently prescribed antibiotics were analyzed to compare their use in the two hospitals. Ceftriaxone and metronidazole did not show any statistically significant differences, while the use of vancomycin, amoxicillin/clavulanic acid, and cefuroxime showed statistically significant differences between the hospitals (Table 4).

Fig 1 shows that the antibiotics cost per prescription (ACPP) in the two hospitals were not normally distributed. So, we used the Mann-Whitney test to compare the results and found that the cost of antibiotics per prescription was significantly different between the two hospitals ($p < 0.001$); it was much higher in the Palestinian Medical Complex, median ($Q_1 - Q_3$) = 592 (198.8–1,409) ILS, while the median ($Q_1 - Q_3$) in Alia Hospital was 187 (94.3–452.3) ILS.

Furthermore, the Palestinian Medical Complex had the highest antibiotic cost per prescription (19,030 ILS). The three highest cost values in the Palestinian Medical Complex were 19,030 ILS, 15,631 ILS, and 7,115 ILS, respectively, and in Alia Hospital, 12,446 ILS, 12,155 ILS, and 9,233 ILS, respectively.

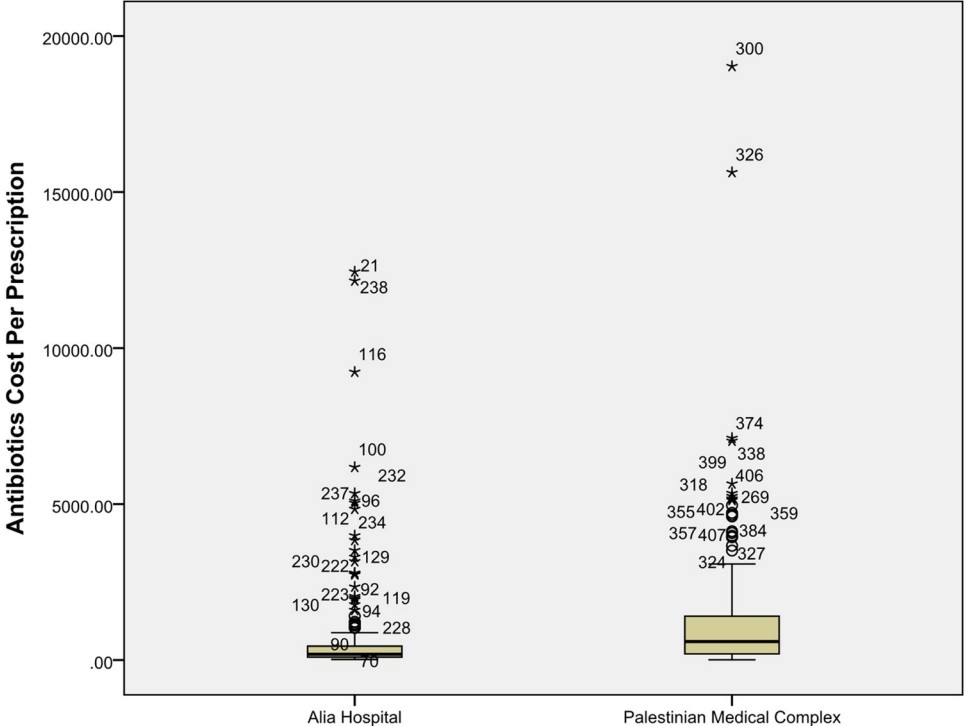

**Fig 1. Distribution of antibiotic cost per prescription in the two hospitals.**

## Discussion

Evaluation of the prescriptions contributes to conducting effective strategies for the elimination of irrational use of antibiotics-related problems. This is the first study that did an in-depth analysis of the antibiotic utilization by two major governmental hospitals in Palestine. Data for this study (428 prescriptions from these hospitals in the West Bank) was collected and analyzed retrospectively. The mean age of patients identified in this study was 40.45 ± 25.47 years.

The current study revealed that males (51.6%) and females (48.4%) were almost equally likely to receive prescriptions for antibiotics. Other studies suggest that female patients may be receiving more antibiotics than male patients [17].

This study reflects the antibiotic prescribing attitudes of almost all groups of physicians, given the fact that 21.3% of the prescriptions were from internal medicine specialists and general practitioners (GPs) (19.4%), and most of the prescriptions were prescribed for intra-abdominal and cardiac infections (18.9%) and as prophylactic to prevent infections (16.1%). A point prevalence survey of antibiotic use in 18 hospitals in Egypt reported that antibiotics were mostly prescribed for surgical and medical prophylaxis [18], and two studies, one a point prevalence survey of antibiotic use in 26 Saudi hospitals in 2016 and another in Turkey to assess antibiotic prescribing at different hospitals and primary health care facilities, showed that the most frequent indication was respiratory tract infections [19,20]. Most likely, this is affected by the hospital wards included.

The average number of drugs per prescription (NDPP) is one of the drug utilization indicators [21]. Our findings suggest that the NDPP was 6.72 ± 4.37, which was higher than the optimal value (1.6–1.8) according to the WHO in collaboration with the International Network of Rational Use of Drugs (INRUD) prescribing indicators. There was considerable over-prescribing of antibiotics, with the number of antibiotics per prescription (NAPP) of 2.61 ± 1.54 and a percentage of 38.9%, while the optimal value is (20.0–26.8%), [22,23]. Moreover, the rate of antibiotic prescribing in Palestine is less than that in some developing countries such as Egypt (59%), Jordan (78.2%), Northern Ireland (46.2%), Iran (68.2%), Latin American countries (54.6%), Kenya (84.8%), Turkey and Tunisia [18,24–28]. This may be connected to doctors' tendency for polypharmacy as well as certain patients' medications being written for many diagnoses. For instance, a widespread practice of prescribing analgesics, respiratory system medications, etc. in addition to antibiotics for infections may also be a contributing factor that affects the percentage of antibiotics prescribed [20]. Thus, since the antibiotic prescribing rate is high in the inpatient settings in Palestine, applying interventions to improve that is necessary.

This study also indicates that the NAPP is affected by other variables: the name of healthcare facility ($p = 0.022$), the diagnosis on prescriptions ($p = 0.006$), the duration of antibiotic therapy ($p < 0.001$), and the number of drugs per prescription ($p < 0.001$). The diagnosis is likely to affect number of antibiotics because some infections require more than one antibiotic according to treatment guidelines to cover the likely pathogens. It is logical that an increased duration of hospitalization is associated with a higher number of antibiotics because that means the case is complicated or not responding. Also it is expected that higher numbers of medications are likely to include higher number of antibiotics.

Due to the COVID-19 pandemic, antibiotic prescriptions for both inpatients and outpatients have increased recently; 8% of COVID-19 patients had bacterial or fungal coinfection, according to a study, despite the fact that 72% of COVID-19 patients are getting broad-spectrum antibiotic therapy [29]. In our study, 12.4% of antibiotic prescriptions were for the treatment of COVID-19 (the SARS coronavirus). This increase in consumption of antibiotics may cause a disastrous effect on resistance in the future.

However, given that 94.63% of antibiotics prescribed were in injectable form, this value was significantly higher than the optimal value (13.4–24.1%), this is much higher than other studies conducted in Ethiopia (22.39%), Indonesian hospitals (85%) [30,31], and among children in Asia (88%), Latin America (81%), and Europe (67%) [32]. Our findings support the view that physicians, in general, are inclined to prescribe parenteral antibiotics, especially for unconscious cases and in-patients. However, frequent injection use increases the risk of contracting blood-borne infections [21], and injections are always more expensive than a comparable oral formulation [33].

It is highly recommended to prescribe oral antibiotics if the case of the patient allows this or to change parenteral medications to oral when the patient's condition is better. It seems that intervention and educational programs are needed to encourage prescribers to use oral medications and antibiotics for inpatients according to the clinical condition and severity.

The most common antibiotic classes prescribed in inpatient settings were "other beta-lactam antibacterials" (the cephalosporins), "other antibacterials" (glycopeptide antibacterials, polymyxins, imidazole derivatives, etc.) and "beta-lactam antibacterials" (penicillins). Furthermore, the most frequently prescribed antibiotics were ceftriaxone, metronidazole, and vancomycin, while the most commonly prescribed antibiotics in the surgical units of two Palestinian hospitals in 2012 were metronidazole, cefuroxime, and ceftriaxone [34].

In 76 countries between 2000 and 2015, penicillins, cephalosporins, quinolones, and macrolides were the most frequently used antibiotic classes [27]. There are some variations in the consumption of different antibiotic classes, despite the fact that broad-spectrum penicillins, carbapenems, and polymyxins consumption has increased in high-, middle-, and low-income countries. For instance, usage of cephalosporins has increased in low-and middle-income countries while declining in high-income countries [27]. Also, in 41 countries in 2012, the most commonly prescribed antibiotics were very broad-spectrum ceftriaxone, cefepime, and meropenem [32].

In Saudi Arabia, the most commonly prescribed antibiotic group was third-generation cephalosporins [19]. Similarly, another study conducted in Nigeria reported the same findings that the most frequently used antibiotics include third generation cephalosporins (mainly ceftriaxone) followed by a combination of penicillins (mainly amoxicillin with enzyme inhibitor) and fluoroquinolones [35]. In Latin American countries, a study analyzing the use of antibiotics indicated that third-generation cephalosporins were the class of antibiotics most frequently used followed by carbapenems and fluoroquinolones [25]. In China, the most prescribed antibiotics groups were cephalosporins, macrolides and fluoroquinolones [36]. In Tanzania, the classes of antibiotics commonly prescribed were penicillins (amoxicillin), fluoroquinolones (ciprofloxacin), and nitroimidazoles (metronidazole) in that order [37]. In Pakistan, ceftriaxone, metronidazole and cefotaxime were the top most frequently prescribed antimicrobials [38].

The mean duration of the antimicrobial treatment prescription was 7.33 ± 8.19 days, similar to a study conducted in Turkey (7 days) [20] and another one in France (8 days) [39], while the mean duration of antimicrobial treatment prescription in India was 5.24 days [40]. According to a survey of American adults, the length of antibiotic therapy should be determined by a validated indicator of clinical stability (resolution of vital sign abnormalities, ability to eat, and normal mentation), and antibiotic therapy should be continued in general for at least 5 days after the patient reaches stability [41]. The duration of antibiotic treatment varies according to the severity of the disease and the nature of the drug. Since there is no consensus on the optimal duration of therapy for the majority of infectious diseases, it is better to treat for at least 7–10 days. A short course of treatment may lead to antimicrobial-resistant microbes. At the same time, prolonged exposure increases the risk of adverse drug reactions, antimicrobial resistance, and also unwanted expenditure on antibiotics [40].

The cost of the prescribed medications is another key drug utilization indicator that is relevant and is thus used to evaluate the performance of the physicians' rational drug use [42]. However, comparisons of antimicrobial agent utilization costs globally could be misleading because of the huge variations in the pricing of drugs. Our study indicated that the average cost per prescription (CPP) was equivalent to 392 ± 744 USD, and the average antibiotic cost per prescription (ACPP) was equivalent to 276 ± 553 USD, which constituted 70.53% of the total cost of all prescribed drugs. While in the Emergency Department of a tertiary care hospital in Saudi Arabia, the average cost of prescribed antibiotics was equivalent to 17.8 ± 11.6 USD only [43]. Another study in the medical intensive care unit (ICU) of a tertiary care teaching hospital in Nepal showed that the average cost of antibiotics per patient was 47.67 ± 63.73 USD [44]. A study in India reported that the average cost of antibiotics per patient in the ICU was 32.58 USD [45]. In 2012, the total cost of antibiotic use over a one-month period in two Palestinian hospitals' surgical units was about 6,300 USD [34]. However, on reviewing data on cost analysis from developed countries, it is found that the ICU antimicrobial agent costs per patient-day varied from 208 USD to 312 USD [46]. It is very important to consider the cost of medications and to choose cheaper antibiotics if they cover the suspected microorganism or if results of culture and sensitivity show that they are enough.

In a comparison of the predominant infections between the two hospitals, they were statistically significantly different ($p < 0.001$). Intra-abdominal and cardiac infections and COVID-19 (the SARS coronavirus) were predominant at the Palestinian Medical Complex, and labor or gynecological infections and using antibiotics for surgical prophylaxis were predominant at Alia Hospital. Furthermore, the use of vancomycin, amoxicillin and clavulanic acid, and cefuroxime were statistically significantly different between the hospitals. The final comparison was the cost of antibiotics per prescription, and it was significantly different between the two hospitals ($p < 0.001$); it was much higher in the Palestinian Medical Complex. The differences in prescribing and cost may be explained by the differences in types of infections and reasons of antibiotic use. However, a more detailed review is needed to find the reasons of this high use and cost of antibiotics at the Palestinian Medical Complex.

Different variables, such as a lack of suitable drug use regulations, protocols, recommendations, and formulary books, may be responsible for the antibiotic usage pattern in this study. Inappropriate antibiotic monitoring and evaluation, microbial resistance, a lack of continuing medical education, polypharmacy, and a lack of clinical pharmacologists or clinical pharmacists are some of the other contributing factors that may result in the overuse and abuse of antibiotics in hospitals [47].

The strength of the study is in the analysis of the cost of medications and antibiotics which is done -to the best of our knowledge- for the first time in our country.

There are some limitations of this study. We explored the antibiotic utilization pattern over a period of 4 months; hence, the influence of seasonal variations on disease patterns and antibiotic utilization could not be considered. Our findings could not be generalized to the whole of Palestine and should not be extrapolated to the international environment. Indeed, antibiotic prescribing can be influenced by many factors, e.g. patient case-mix, prevalence of different types of infections, AMR patterns, and institutional factors. The findings do, however, add to a growing literature, particularly around medicine use and pharmaceutical health systems in developing countries.

## Conclusions and recommendations

This study made a detailed assessment of the prescriptions containing antibiotics prescribed in hospitals in Palestine and gave some interesting findings about antibiotics. It has been

concluded that the utilization of antibiotics and their cost in the public hospitals of Palestine are very high and potentially inappropriate. The most commonly prescribed antibiotic classes are the broad spectrum ones. The most commonly prescribed antibiotic was ceftriaxone, and the most commonly encountered infections in inpatients were intra-abdominal infections. Intra-abdominal and cardiac infections and COVID-19 (the SARS coronavirus) were predominant at the Palestinian Medical Complex, and labor or gynecological infections and using antibiotics for surgical prophylaxis were predominant at Alia Hospital. Furthermore, the use of vancomycin, amoxicillin and clavulanic acid, and cefuroxime were statistically significantly different between the hospitals. The cost of antibiotics per prescription was significantly different between the two hospitals; it was much higher in the Palestinian Medical Complex.

The results of the study support the suggestion that continuous training and education programs for medical professionals about the rational use of antibiotics and injections and their subsequent pharmacoeconomic evaluation be established and monitored in order to make the necessary adjustments in prescribing sustainable. A feedback monitoring system for physicians' antibiotic prescriptions will greatly enhance their prescribing practices. Knowledge and compliance with updated clinical guidelines are also recommended to enhance rational prescribing. Having clinical pharmacists and infectious disease specialists in the wards may help.

## Supporting information

**S1 Table. Minimal data set.**
(XLSX)

## Acknowledgments

The authors would like to thank the Palestinian Ministry of Health and the administration of the hospitals for their permission to conduct the study.

## Author Contributions

**Conceptualization:** Rufayda Dawood Manassrah, Rowa Al Ramahi.

**Data curation:** Rufayda Dawood Manassrah.

**Formal analysis:** Rufayda Dawood Manassrah.

**Investigation:** Rufayda Dawood Manassrah.

**Methodology:** Rufayda Dawood Manassrah.

**Project administration:** Rufayda Dawood Manassrah, Rowa Al Ramahi.

**Resources:** Rufayda Dawood Manassrah.

**Software:** Rufayda Dawood Manassrah.

**Supervision:** Rowa Al Ramahi.

**Validation:** Rufayda Dawood Manassrah, Rowa Al Ramahi.

**Visualization:** Rufayda Dawood Manassrah, Rowa Al Ramahi.

**Writing – original draft:** Rufayda Dawood Manassrah.

**Writing – review & editing:** Rowa Al Ramahi.

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
