## [Decision Letter · Decision Letter 0]

20 Feb 2024

PONE-D-24-01778Assessment of antibiotic prescribing pattern and cost for hospitalized patients: a study from PalestinePLOS ONE

Dear Dr. Manassrah,

Thank you for submitting your manuscript to PLOS ONE. After careful consideration, we feel that it has merit but does not fully meet PLOS ONE’s publication criteria as it currently stands. Therefore, we invite you to submit a revised version of the manuscript that addresses the points raised during the review process.

We look forward to receiving your revised manuscript.

Kind regards,

Chikezie Hart Onwukwe

Academic Editor

PLOS ONE

6. Please include a new copy of Table 2 in your manuscript; the current table is difficult to read. Please follow the link for more information: https://blogs.plos.org/plos/2019/06/looking-good-tips-for-creating-your-plos-figures-graphics/

Reviewers' comments:

Reviewer's Responses to Questions

**Comments to the Author**

1. Is the manuscript technically sound, and do the data support the conclusions?

Reviewer #1: Yes

Reviewer #2: Yes

2. Has the statistical analysis been performed appropriately and rigorously? 

Reviewer #1: I Don't Know

Reviewer #2: Yes

3. Have the authors made all data underlying the findings in their manuscript fully available?

Reviewer #1: Yes

Reviewer #2: No

4. Is the manuscript presented in an intelligible fashion and written in standard English?

Reviewer #1: Yes

Reviewer #2: Yes

5. Review Comments to the Author

Reviewer #1: Thank you for your research in this topic.

Only inpatient prescriptions that included antibiotics for any reason were collected from the computerized system (Avicenna HIS) , My concern is about the ratio of prescriptions that are not included in the computerized system, papers written without digital data logging?

Reviewer #2: In general, it is a good written manuscript. The manuscript presented in clear and comprehensible fashion and written in standard English. However, for further improvement and to shape up the manuscript, I wish to point out some modifications: the alignment of Tables 2 and 3 should be adjusted.

6. PLOS authors have the option to publish the peer review history of their article (what does this mean?). If published, this will include your full peer review and any attached files.

Reviewer #1: **Yes: **Ahmad Al Athamneh

Reviewer #2: **Yes: **Dr. Thaer Abdelghani

---

## [Author Response · Author response to Decision Letter 0]

2 Apr 2024

We would like to thank the reviewers for their time and useful comments.

---

## [Decision Letter · Decision Letter 1]

15 Apr 2024

Assessment of antibiotic prescribing pattern and cost for hospitalized patients: a study from Palestine

PONE-D-24-01778R1

Dear Dr. Rufayda Dawood Manassrah,

We’re pleased to inform you that your manuscript has been judged scientifically suitable for publication and will be formally accepted for publication once it meets all outstanding technical requirements.

Kind regards,

Chikezie Hart Onwukwe

Academic Editor

PLOS ONE

Additional Editor Comments (optional):

Reviewers' comments:

Reviewer's Responses to Questions

**Comments to the Author**

1. If the authors have adequately addressed your comments raised in a previous round of review and you feel that this manuscript is now acceptable for publication, you may indicate that here to bypass the “Comments to the Author” section, enter your conflict of interest statement in the “Confidential to Editor” section, and submit your "Accept" recommendation.

Reviewer #2: All comments have been addressed

2. Is the manuscript technically sound, and do the data support the conclusions?

Reviewer #2: Yes

3. Has the statistical analysis been performed appropriately and rigorously? 

Reviewer #2: Yes

4. Have the authors made all data underlying the findings in their manuscript fully available?

Reviewer #2: Yes

5. Is the manuscript presented in an intelligible fashion and written in standard English?

Reviewer #2: Yes

6. Review Comments to the Author

Reviewer #2: The manuscript technically sound, and the data support the conclusions. The statistical analysis has been performed appropriately. The conclusions drawn appropriately based on the data presented. The manuscript presented in clear and comprehensible fashion and written in standard English.

7. PLOS authors have the option to publish the peer review history of their article (what does this mean?). If published, this will include your full peer review and any attached files.

Reviewer #2: **Yes: **Dr. Thaer Abdelghani

---

## [Editor Report · Acceptance letter]

24 Apr 2024

PONE-D-24-01778R1 

PLOS ONE

Dear Dr. Manassrah, 

I'm pleased to inform you that your manuscript has been deemed suitable for publication in PLOS ONE. Congratulations! Your manuscript is now being handed over to our production team.

Kind regards, 

on behalf of

Dr. Chikezie Hart Onwukwe 

Academic Editor

PLOS ONE